# Targeting Oncogenic KRAS in Non-Small-Cell Lung Cancer

**DOI:** 10.3390/cancers13235956

**Published:** 2021-11-26

**Authors:** Noriaki Sunaga, Yosuke Miura, Norimitsu Kasahara, Reiko Sakurai

**Affiliations:** 1Department of Respiratory Medicine, Gunma University Graduate School of Medicine, 3-39-15 Showa-machi, Maebashi 371-8511, Gunma, Japan; m15702011@gunma-u.ac.jp; 2Innovative Medical Research Center, Gunma University Hospital, 3-39-15 Showa-machi, Maebashi 371-8511, Gunma, Japan; m14702016@gunma-u.ac.jp; 3Oncology Center, Gunma University Hospital, 3-39-15 Showa-machi, Maebashi 371-8511, Gunma, Japan; m15702012@gunma-u.ac.jp

**Keywords:** v-Ki-ras2 Kirsten rat sarcoma viral oncogene, non-small-cell lung cancer, covalent KRAS G12C inhibitor, drug resistance, combination therapy

## Abstract

**Simple Summary:**

v-Ki-ras2 Kirsten rat sarcoma viral oncogene (*KRAS*) is the most common driver in NSCLC, and targeting oncogenic KRAS is a major challenge in the treatment of non-small-cell lung cancer (NSCLC). While several covalent KRAS G12C inhibitors have emerged as a novel anti-KRAS therapy, the development of combined therapies involving the targeting of oncogenic KRAS plus other targeted drugs is still required given the vast heterogeneity of *KRAS*-mutated tumors. In this review, we summarize the biological and immunological characteristics of oncogenic *KRAS*-driven NSCLC and the preclinical and clinical evidence for mutant KRAS-targeted therapies. We also discuss the mechanisms of resistance to KRAS G12C inhibitors and possible therapeutic strategies to overcome this drug resistance.

**Abstract:**

Recent advances in molecular biology and the resultant identification of driver oncogenes have achieved major progress in precision medicine for non-small-cell lung cancer (NSCLC). v-Ki-ras2 Kirsten rat sarcoma viral oncogene (*KRAS*) is the most common driver in NSCLC, and targeting KRAS is considerably important. The recent discovery of covalent KRAS G12C inhibitors offers hope for improving the prognosis of NSCLC patients, but the development of combination therapies corresponding to tumor characteristics is still required given the vast heterogeneity of *KRAS*-mutated NSCLC. In this review, we summarize the current understanding of *KRAS* mutations regarding the involvement of malignant transformation and describe the preclinical and clinical evidence for targeting *KRAS*-mutated NSCLC. We also discuss the mechanisms of resistance to KRAS G12C inhibitors and possible combination treatment strategies to overcome this drug resistance.

## 1. Introduction

Lung cancer is the leading cause of cancer-related death worldwide. Lung cancer is classified into two major histological subtypes: small-cell lung cancer (SCLC) and non-small-cell lung cancer (NSCLC), with the latter accounting for approximately 85% of all lung cancers. NSCLC consists mainly of lung adenocarcinomas (LUADs) and squamous cell lung carcinomas. The prognosis of lung cancer has been poor because the majority of lung cancer patients are initially diagnosed at an advanced stage. While recent developments in molecularly targeted drugs and immune checkpoint inhibitors have prolonged the survival of patients with advanced NSCLC [1,2], v-Ki-ras2 Kirsten rat sarcoma viral oncogene (*KRAS*), which is one of the most common oncogenes in NSCLC, had long been an “undruggable target” despite extensive efforts. RAS proteins are small guanosine triphosphatase (GTP)-binding proteins. The RAS family consists of three members, HRAS, NRAS and KRAS. *KRAS* knockout in mice is lethal, but *HRAS* or *NRAS* knockout mice are viable and develop normally [3,4], and KRAS can be replaced by HRAS during mouse embryogenesis [5]. Upon activation of receptor tyrosine kinases (RTKs), RAS is activated by SHP2 in cooperation with the adaptor protein GRB2 and SOS, which is a guanine nucleotide exchange factor (GEF) that catalyzes the transition of the guanosine diphosphate (GDP)-bound inactive form of RAS to the GTP-bound active form [6,7]. When *KRAS* is mutated, RAS is locked into the GTP-bound active form, which in turn constitutively activates downstream signaling pathways, such as the RAF-MEK-ERK and PI3K-AKT-mTOR pathways, conferring malignant phenotypes [8,9]. Recently, several covalent KRAS G12C inhibitors, including AMG510 (sotorasib) and MRTX849 (adagrasib), have been developed for *KRAS* G12C-mutant tumors [10,11,12]. Of note, the CodeBreaK100 clinical trial showed a beneficial effect of sotorasib in patients with advanced NSCLC harboring *KRAS* G12C mutations [13], and the FDA approved sotorasib for *KRAS* G12C-positive NSCLC patients who had received at least one prior systemic therapy in May 2021. On the other hand, KRAS G12C inhibitors do not provide as durable responses as tyrosine kinase inhibitors against EGFR or ALK; the median progression-free survival times of NSCLC patients treated with sotorasib and the EGFR-tyrosine kinase inhibitor (EGFR-TKI) osimertinib were 6.8 months [13] and 18.9 months [14], respectively. Furthermore, several secondary *KRAS* mutations and other molecular abnormalities causing resistance to covalent KRAS G12C-specific inhibitors have been reported [15,16,17]. The precise resistance mechanisms and therapeutic strategies for KRAS G12C inhibitor-resistant tumors are under investigation [18]. In this review, we summarize the current understanding of oncogenic *KRAS* mutations in terms of the biological and immunological characteristics that impact the malignant phenotypes of NSCLC, and describe the preclinical and clinical evidence for targeting mutant KRAS. We also discuss the mechanisms of resistance to KRAS G12C inhibitors, and possible therapeutic strategies to overcome this drug resistance.

## 2. Genetic Alterations of *KRAS* in Human Cancers

Mutations in members of the *RAS* gene family are commonly found in various human cancers; however, the mutation frequencies of RAS isoforms differ among human cancers. *KRAS* is the predominantly mutated member (85%), followed by *NRAS* (11%) and *HRAS* (4%) [19]. *KRAS* mutations are commonly detected in pancreatic cancer, colorectal cancer and NSCLC [20,21]. *KRAS* mutations mainly occur at codons 12, 13 and 61, and the G12C mutation is the most common (41%) in NSCLC, whereas the *KRAS* G12D and G12V mutations are the major subtypes in colorectal and pancreatic cancers (Figure 1) [22]. Thus, the profiles of *KRAS* mutation subtypes differ according to the type of human cancer.

*KRAS* copy number alterations have been reported in NSCLC [23]. A mutant allele-specific imbalance (MASI) in *KRAS* frequently occurred, and MACI with *KRAS* copy number gain was related to increased RAS GTPase activity [24]. *KRAS* MASI has been associated with an unfavorable prognosis in LUAD [24,25,26]. In a preclinical study, the mutant *KRAS* copy number gain resulted in metabolic rewiring and increased malignancy [27]. *KRAS* amplification was also shown to be a mechanism of resistance to EGFR-TKIs [28]. These studies suggest that *KRAS* copy number alterations enhance oncogenic KRAS activity, leading to treatment resistance and a worse prognosis in *KRAS*-mutated NSCLC.

## 3. Multiple Faces of *KRAS*-Mutated NSCLC

The biological and clinical significance of *KRAS* mutations varies depending on the mutation subtype in NSCLC. For instance, G12C and G12D have higher affinities for binding to RALGDS and PI3K, respectively [29]. A recent study assessing the molecular profiles of NSCLC specimens revealed that tumors harboring *KRAS* G12C mutations exhibited a higher programmed death-ligand 1 (PD-L1) tumor proportion score and tumor mutational burden than those with *KRAS* non-G12C mutations [30]. From a clinical point of view, NSCLC patients harboring *KRAS* G12C or G12V mutations have a worse prognosis than those without these mutation subtypes [29]. In another study, surgically resected LUAD patients with *KRAS* G12C mutations had significantly shorter survival times than those with *KRAS* non-G12C mutations [31]. Therefore, targeting *KRAS* G12C seems to be an attractive therapeutic approach to improve the prognosis of NSCLC patients. In contrast, no significant difference in the overall survival of patients with metastatic NSCLC was found among different subtypes of *KRAS* mutations [32,33]. These inconsistent observations are likely due to the intratumoral heterogeneity of *KRAS*-mutated tumors [34,35], in addition to biological and immunological differences in *KRAS* mutation subtypes [29,36].

Omics data analyses have attempted to classify *KRAS*-mutated NSCLCs into subgroups with distinct biological and immunological characteristics to predict therapeutic vulnerabilities. Skoulidis et al. conducted an integrative analysis of genomic, transcriptomic and proteomic data, and defined three subtypes of *KRAS*-mutated LUADs based on co-occurring genetic events: the KC subgroup was defined by *CDKN2A/B* inactivation and reduced TTF-1 expression, the KL subgroup was defined by *LKB1* alterations, and the KP subgroup was defined by *TP53* alterations [37]. KC tumors exhibited gastrointestinal-like differentiation with upregulated biomarkers for mucinous differentiation, such as CK20 and MUC5B, and reduced mTOR signaling. KL tumors had high rates of mutations in the *KEAP1* gene encoding Kelch-like ECH-associated protein 1, and expressed lower levels of immune markers, such as lower PD-L1 levels, whereas KP tumors showed higher levels of somatic mutations, inflammatory markers, and immune checkpoint effector molecules, including PD-L1 and programmed death 1 (PD-1), accompanied by increased T cell infiltration [37]. As described above, the *KEAP1*-inactivating mutation is one of the more frequently co-occurring genetic alterations in *KRAS*-mutant/*LKB1*-mutant LUADs [37,38]. KEAP1 is a negative regulator of the transcriptional factor NRF2 [39], and *KEAP1* inactivation aberrantly activates the NRF2 pathway, conferring a more aggressive tumor phenotype [40,41,42]. The inactivating mutations in *KEAP1* and *LKB1*, as well as the activation of the YAP1 and/or RSK-mTOR pathway, contribute to mutant KRAS independency in tumor cells, potentially causing therapeutic resistance to KRAS inhibitors [43]. Thus, coexisting genetic events may be associated with therapeutic vulnerabilities in *KRAS*-mutated NSCLC.

## 4. Oncogenic KRAS Regulates the Tumor Microenvironment (TME)

There is growing evidence that oncogenic KRAS is involved in tumor immune evasion by regulating the TME [44]. *KRAS*-mutated NSCLC tumors have inflammatory characteristics; oncogenic KRAS induces several inflammatory cytokines and chemokines, including IL-6, IL-8, CXCL1 and CCL5, which influence the TME [45]. For instance, oncogenic KRAS mediates the secretion of IL-6, which drives protumor M2-type macrophage polarization along with the recruitment of myeloid-derived suppressor cells (MDSCs) in lung tumors [46]. Oncogenic KRAS also upregulates the expression of IL-8, a neutrophil chemoattractant, through MEK-ERK pathway activation in NSCLC cells [47]. Elevated serum IL-8 levels were found to be associated with an unfavorable prognosis in NSCLC patients treated with the PD-l antibody nivolumab, and increased IL-8 expression in tumors was negatively correlated with T cell markers and IFNγ-dependent gene signatures in the TME [48]. Another neutrophil chemoattractant, CXCL1, is also upregulated by *KRAS* mutations, and is involved in the development of lung cancer [49,50,51]. Moreover, *KRAS*-mutated NSCLC cells produce CCL5 [52], which plays an important role in antitumor immunity by promoting the recruitment of T cells and dendritic cells to the TME [53].

Several studies have demonstrated that *KRAS*-mutated NSCLC tumors exhibit PD-L1 overexpression [54], which is induced by oncogenic RAS-related pathway activation [55,56,57,58]. A previous study showed that B7-H3, T-cell immunoglobulin mucin family member 3, and indoleamine 2,3-dioxygenase-1 were highly expressed in tumor stroma-associated inflammatory cells in LUADs harboring *KRAS* mutations [59]. In addition to immune checkpoint molecules, regulatory T cells (Tregs) play a pivotal role in tumor immune evasion [60]. Mutant KRAS induces Tregs via secretion of IL-10 and transforming growth factor-β1 from tumor cells [61]. We previously identified *NT5E*, encoding CD73, as a gene upregulated by *KRAS* mutations in NSCLC cells [62]. Similarly, Le et al. found CD73 overexpression in *EGFR*-mutated NSCLC tumors [63]. They further demonstrated that the proportion of Tregs was decreased by coculturing these cells with conditioned medium from *EGFR*-mutated NSCLC cells with CD73 knockdown, and that an anti-CD73 antibody suppressed tumor growth in immunocompetent mice [63]. These observations suggest that *KRAS*-mutated NSCLC tumors have the ability to evade immune responses by regulating inflammatory cytokines and chemokines, immune checkpoint molecules and Tregs. Thus, targeting mutant KRAS may be an optional approach to abolish tumor immune evasion, and combination treatments including KRAS G12C inhibitors and immune checkpoint inhibitors could be effective therapeutic strategies.

Genetic alterations of *TP53*, *LKB1* or *KEAP1* co-occurring with *KRAS* mutations affect the TME. In a preclinical study, the ectopic expression of LKB1 induced a strong increase in PD-L1 expression in human NSCLC cells harboring *KRAS* mutations, whereas LKB1 knockdown decreased PD-L1 in mouse lung fibroblasts transformed with active KRAS and HPV16 E6E7 [64]. *LKB1* mutations were negatively associated with PD-L1 expression in *KRAS*-mutated NSCLC tumors, whereas gene expression signatures associated with the infiltration of effector T cells and dendritic cells were reduced in LUAD tumors with *LKB1* mutations and NRF2 activation [42]. Meanwhile, *KRAS*/*TP53*-comutated LUAD tumors manifested increased expressions of PD-1 and PD-L1, a high accumulation of CD8^+^ tumor-infiltrating lymphocytes and high mutational loads, and NSCLC patients with co-occurring *KRAS*/*TP53* mutations showed a favorable clinical benefit with anti-PD-1 therapy [65].

Vascular endothelial growth factor (VEGF) is a key player in tumor-induced immunosuppression in the TME [66,67,68]. VEGF prevents dendritic cell development [69] and enhances the coexpression of inhibitory receptors involved in CD8^+^ T cell exhaustion in tumors, including PD-1 and cytotoxic T-lymphocyte antigen 4 (CTLA-4) [70]. VEGF also induces the accumulation of Tregs and MDSCs in tumors [69,71]. The relationship between VEGF and KRAS has been identified in previous studies, which showed that VEGF was upregulated by oncogenic KRAS [72,73] and that VEGF promoter activity was upregulated by activation of the PI3K-AKT signaling pathway [74]. Of note, a recent clinical study demonstrated that the addition of the anti-VEGF antibody bevacizumab to a regimen including the anti-PD-L1 antibody atezolizumab plus carboplatin and paclitaxel (ABCP) produced longer progression-free survival and a greater overall survival benefit than atezolizumab plus carboplatin and paclitaxel (ACP), or bevacizumab plus carboplatin and paclitaxel (BCP), in patients with metastatic nonsquamous NSCLC with *KRAS* mutations [75]. Moreover, a greater survival benefit was achieved with ABCP compared with ACP or BCP in patients with tumors carrying *KRAS*/*LKB1*/*KEAP1* comutations. These observations suggest the therapeutic significance of anti-VEGF therapies in combination with immune checkpoint inhibitors in patients with NSCLC harboring *KRAS* mutations.

## 5. Covalent KRAS G12C Inhibitors for *KRAS*-Mutated NSCLC

The recent discovery of covalent KRAS G12C inhibitors has changed “undruggable” oncogenic KRAS into a “druggable” target. In the KRAS G12C inactive (GDP-bound) form, the mutant cysteine residues adjacent to the switch II pocket are involved in the effector interaction [76]. The binding of covalent KRAS G12C inhibitors, such as sotorasib and adagrasib, to the switch pocket changes the nucleotide preference to favor GDP, thus impairing oncogenic KRAS-mediated signal transduction and leading to tumor regression in preclinical models [10,11]. The phase 1 portion of the CodeBreak100 clinical trial for patients with pretreated advanced solid tumors carrying *KRAS* G12C mutations demonstrated the encouraging safety and anticancer activity of sotorasib monotherapy [77]. Subsequently, the phase 2 portion of the CodeBreak100 trial evaluated the efficacy and safety of sotorasib, administered orally at 960 mg once daily, in *KRAS* G12C-mutant NSCLC patients who had received standard therapies [13]. Of the 126 enrolled patients, the majority had a history of smoking (92.9%) and a history of treatment with both platinum-based chemotherapy and inhibitors of PD-1 or PD-L1 (81.0%). The objective response rate and disease control rate were 37.1% and 80.6%, respectively. The median duration of response, median progression-free survival time and median overall survival time were 11.1 months, 6.8 months and 12.5 months, respectively. Treatment-related adverse events were observed in 88 patients (69.8%), including grade 3 events in 25 patients (19.8%) and grade 4 events in one patient (0.8%). Based on the above clinical efficacy and safety data, sotorasib was approved by the FDA as the first mutant KRAS-targeted drug for *KRAS* G12C-mutated advanced or metastatic NSCLC patients who have received at least one prior systemic therapy. Considering that the standard first-line therapy for patients with advanced NSCLC is an anti-PD-1/L1 antibody in combination with platinum-based chemotherapy [78], it is plausible that *KRAS* G12C-mutated NSCLC patients generally receive sotorasib in the second-line setting after chemotherapy with immune checkpoint inhibitors. It is noteworthy that a subgroup of patients treated with prior anti-PD-1/L1 therapy but not platinum-based chemotherapy had a high objective response rate (69.2%) and a median overall survival time of 17.7 months [79]. Similarly, the KRYSTAL-1 phase 1/2 trial evaluating adagrasib in patients with advanced or metastatic solid tumors, including *KRAS* G12C-mutant NSCLC patients who were previously treated with chemotherapy and an anti-PD-1/L1 antibody, showed that 45% of the patients (23/51) achieved a partial response (PR) and 26 patients had stable disease (SD) [80]. In this trial, the levels of transcripts of immune molecules, such as CD4 and CD8, were increased after treatment with adagrasib in patients with NSCLC harboring *KRAS* G12C and *LKB1* coalterations. These results suggest that NSCLC patients are sensitized to immune checkpoint inhibitors by the KRAS G12C inhibitors through tumor immune microenvironment reconditioning, as shown in preclinical models [10,81].

## 6. Resistance Mechanisms for Anti-KRAS-G12C Therapies

While targeted therapies induce a dramatic tumor response early in treatment, most tumors ultimately become resistant to targeted drugs, due to the intratumoral heterogeneity that results in increasing levels of preexisting resistant clones and tumor evolution favoring gain of resistance [82,83]. For *KRAS*-mutated NSCLC, it was shown that co-occurring *KRAS* mutations, including the pairs G12D/G12V, G12D/G12C, G12V/G12C and G12D/G12S, can be found in the same NSCLC tumors [35]. A next-generation sequencing analysis of 1078 patients with NSCLC carrying *KRAS* mutations uncovered that 53.5% of the patients had additional mutations, including *MET* amplifications (15.4%), *ERBB2* amplifications (13.8%), *PIK3CA* mutations (3.2%), *EGFR* mutations (1.2%) and *BRAF* mutations (1.3%) [33]. In another study evaluating concomitant genetic alterations in *KRAS*, *EGFR*, *ALK*, *ROS1* and *BRAF* in 3774 samples from NSCLC patients, the frequencies of single *KRAS* mutations and concomitant *KRAS* mutations were 8.0% and 1.1%, respectively, indicating that approximately 12% of *KRAS*-mutated NSCLC tumors had other driver mutations [84]. Such genetic alterations could cause resistance to anti-KRAS therapies in *KRAS*-mutated tumors.

Recently, several studies have revealed that secondary *KRAS* mutations and activated bypass pathways potentially cause resistance to covalent KRAS G12C inhibitors in NSCLC [18]. *KRAS* mutations (G12D, G12V, G12W, R68S, H95D and Y96C), high-level amplifications of the *KRAS* G12C allele and alterations in RAS-related pathways (*MET* amplification, *BRAF* V600E mutation, *MAPK2K1*/*MEK1* E102_I103 deletion and *PI3KCA* H1047R mutation) were found in tissue biopsy samples and/or circulating tumor DNA obtained from *KRAS* G12C-mutant NSCLC patients who had acquired resistance to adagrasib [16]. In this study, R68S, H95D and Y96C within the switch II site, where sotorasib and adagrasib bind, were found to confer resistance to KRAS G12C inhibitors in in vitro experiments using Ba/F3 cells expressing *KRAS* G12C mutants; G12C/R68S, G12C/H95D and G12C/Y96C double mutants mediated resistance to adagrasib, while only G12C/H95D retained sensitivity to sotorasib. Similarly, Tanaka et al. found 10 distinct mutations (*KRAS*—G12V, G12F, G13D and Y96D; *NRAS*—Q61L, Q61R, and Q61K; *BRAF*—V600E; *MAPK2K1*/*MEK1*—K57N, Q56P, and E102_I103del) that affect RAS-MAPK signaling in cell-free DNA obtained from a patient who had developed progressive disease after adagrasib monotherapy [17]. In that study, G12C/Y96D transduction into cells induced resistance to KRAS G12C inhibitors (sotorasib, adagrasib and ARS-1620) in cell viability assays using the Ba/F3, MIA PaCa-2 and H358 cell lines; however, these resistant cells were found to be sensitive to the KRAS G12C inhibitor RM-018, which forms a binary complex with cyclophilin A. A N-ethyl-N-nitrosourea mutagenesis screen using Ba/F3 cells transduced with *KRAS* G12C mutants revealed several secondary *KRAS* mutations causing resistance to sotorasib and adagrasib [15]. Both Y96D and Y96S mutants were resistant to both sotorasib and adagrasib, whereas the sensitivities of some of the *KRAS* mutants, such as G13D, R68M and A59T, differed depending on the inhibitor. The study also demonstrated that combined treatment with the SOS1 inhibitor BI-3406 plus the MEK inhibitor trametinib effectively reduced the viability of *KRAS* G12C-mutant NSCLC cells, in which the Y96D or Y96S mutant was exogenously introduced. Therefore, in addition to the genetic alterations that reactivate RAS pathways, *KRAS* Y96C/Y96D/Y96S mutations that hamper the interaction with covalent KRAS G12C inhibitors are likely critical for acquired resistance. The resistance mechanisms described above are summarized in Figure 2. Alternatively, exploratory analyses of the CodeBreak100 trial showed that a subset of patients with *KEAP1* mutations were less responsive to sotorasib, irrespective of their *LKB1* mutation status: the response rates for *LKB1*-mutant/*KEAP1*-mutant and wild-type *LKB1*/*KEAP1*-mutant patients were 23% (3/13) and 14% (1/7), respectively [13]. Further biomarker studies with a larger number of patients will elucidate the predictive value of *KEAP1* mutations.

An epithelial-to-mesenchymal cell transition (EMT) program contributes to the therapeutic resistance to various types of anticancer drugs, including EGFR-TKIs [85,86,87,88]. Previous studies have shown that mutant KRAS induces EMT in pancreatic and ovarian cancers [89,90]. Activation of the kinase AXL induces EGFR-TKI resistance associated with the mesenchymal phenotype [87,88], and the dual blockade of MEK and AXL overcomes EMT-mediated drug resistance and synergistically suppresses tumor growth in *KRAS*-mutated lung cancer cells [91]. Additionally, *KRAS* G12C-carrying tumors were shown to express higher levels of genes relevant to EMT [31]. A recent preclinical study demonstrated EMT-induced resistance to sotorasib via IGFR-dependent, KRAS G12C-independent activation of the PI3K-AKT pathway in *KRAS*-mutated NSCLC cells, and a combination treatment with sotorasib plus the SHP2 inhibitor SHP099 and the PI3K inhibitor GDC-0941 effectively impaired the growth of *KRAS* G12C-mutant tumors [92].

## 7. Combined Therapies Involving Targeting of Oncogenic KRAS plus Other Targeted Drugs for *KRAS*-Mutated NSCLC

Oncogenic RAS upregulates downstream pathways and negatively regulates RTK signaling, whereas attenuating oncogenic RAS results in compensatory activation of RTK and wild-type RAS signaling pathways [93]. Thus, mutant RAS and wild-type RAS appear to complement each other to activate RTK-RAS signaling pathways to sustain tumor growth and survival, and the multitargeting of RAS signaling pathways could be an effective therapeutic approach. In our previous work, KRAS mutant-specific knockdown using short hairpin RNA vectors targeting *KRAS* G12C or *KRAS* G12V significantly but incompletely suppressed tumor growth in four NSCLC cell lines, all of which retained the wild-type *KRAS* allele [62]. In that study, mutant KRAS knockdown inactivated the MEK-ERK pathway in a common manner, but resulted in variable changes in RAS-related signaling (e.g., changes in phospho-EGFR levels and phospho-Akt levels), and sensitized *KRAS*-mutated NSCLC cells to p38 and EGFR inhibitors to varying degrees. We also reported that dual p38 and MEK inhibition effectively impaired the tumor growth of *KRAS*-mutated NSCLC cells [94]. Other researchers reported similar results, showing that the dual inhibition of KRAS and p38 inhibited tumor growth in *KRAS*-mutated colorectal cancer cells [95] and *KRAS*-mutated LUAD cells [96]. These findings suggest that blocking p38 or EGFR enhances the growth-inhibitory effect of KRAS G12C inhibitors on NSCLC tumors. Furthermore, the combined inhibition of MEK and PI3K was shown to effectively inhibit the growth of *KRAS*-mutated NSCLC tumors in experimental models [97,98,99]. In another study by Kitani et al., effective combination strategies differed between epithelial-like and mesenchymal-like *KRAS*-mutant NSCLC cells; a combination therapy with the MEK inhibitor trametinib plus the EGFR-TKI afatinib was effective for epithelial-like tumors, while the dual blockade of MEK and FGFR effectively suppressed mesenchymal tumor growth [100]. A recent study demonstrated that combined treatment with the KRAS G12C inhibitor ARS-1620 plus the pan-PI3K inhibitor GDC0941 consistently suppressed tumor growth in ARS-1620-resistant *KRAS*-mutated tumors, whereas the growth-inhibitory effects of ARS-1620 plus afatinib or cetuximab varied among *KRAS*-mutated cell lines [101]. Together, these observations indicate that the therapeutic efficacy of covalent KRAS G12C inhibitors varies depending on the biological properties of the treated *KRAS*-mutated tumors.

Accumulating evidence has provided promising results regarding inhibitors of the nonreceptor protein tyrosine phosphatase SHP2, encoded by *PTPN11*, in cancer [102]. SHP2 is ubiquitously expressed in all human tissues and is required for the RTK-mediated activation of RAS signaling pathways [6]. SHP2 also serves as an immunomodulator that regulates the PD-1 cascade in immune evasion [102]. In mutant *KRAS*-driven murine models of NSCLC, genetic deletion of *PTPN11* suppressed tumor development, while the dual inhibition of SHP2 and MEK resulted in sustained tumor growth control in murine- and human patient-derived organoids and xenograft models of NSCLC [103]. Furthermore, combined treatment with the SHP2 inhibitor TNO155 plus a covalent KRAS G12C inhibitor achieved the effective and sustained inhibition of cell growth and MEK-ERK pathway activation in *KRAS*-mutated NSCLC cells [104]. Considering that SHP2 is associated with most RTKs [105], combination therapy with SHP2 inhibitors plus KRAS G12C inhibitors could be an attractive therapeutic strategy for refractory or resistant NSCLC harboring *KRAS* mutations. The clinical efficacies of combinations of KRAS G12C inhibitors plus SHP2 inhibitors are currently under investigation in clinical trials. Altogether, it is likely that combined therapy, including the targeting of oncogenic KRAS, will be required for effective treatment given the vast intratumoral heterogeneity of *KRAS*-mutated tumors and the involvement of KRAS in numerous effector pathways that mediate tumorigenesis [106]. The ongoing clinical trials assessing combined treatments with covalent KRAG G12C inhibitors plus other anticancer drugs will elucidate effective therapeutic strategies for *KRAS*-mutated NSCLC patients (Table 1).

## 8. Conclusions

The discovery of covalent KRAS G12 inhibitors has produced dramatic advances in the treatment of *KRAS*-mutated NSCLC. Currently, many clinical trials are ongoing to evaluate the efficiencies of KRAS G12 inhibitor monotherapy and its combination with various types of anticancer drugs. Considering the vast heterogeneity of tumors driven by mutant KRAS, it is plausible that covalent KRAS G12C inhibitors will be therapeutically effective when combined with other anticancer drugs. Thus, the battle against oncogenic KRAS, moving toward improvements in the prognosis of NSCLC patients, has just begun.

## Figures and Tables

**Figure 1 cancers-13-05956-f001:**
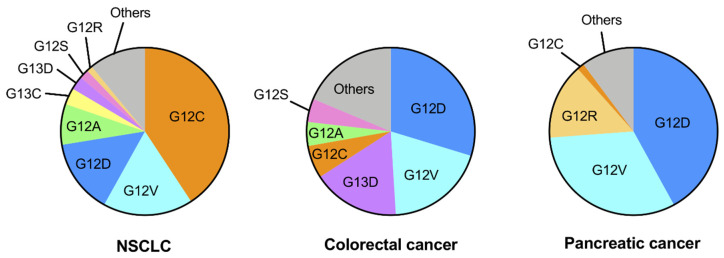
Frequencies of *KRAS* mutation subtypes in NSCLC, colorectal cancer and pancreatic cancer. The pie charts were made based on data from the AACR Project GENIE: Powering Precision Medicine through an International Consortium (GENIE Cohort v10.0-public; Ref. [22]).

**Figure 2 cancers-13-05956-f002:**
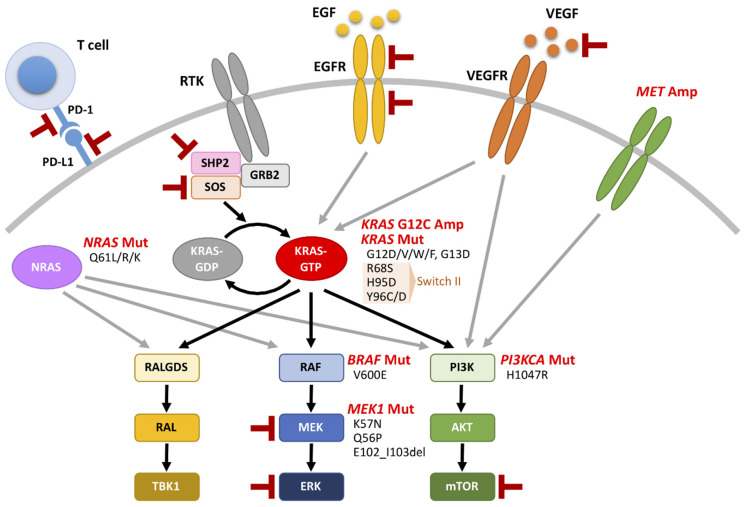
Schematic of Ras signaling pathways and the mechanisms of resistance to covalent KRAS G12C inhibitors in NSCLC. Amplification of the *KRAS* G12C allele, mutations in *KRAS*, *NRAS*, *BRAF*, *MEK1* and *PI3KCA,* and *MET* amplification (marked in red) have been observed in patients with acquired resistance to sotorasib or adagrasib [16,17]. T symbol in red: therapeutic targets combined with KRAS G12 inhibitors in clinical trials (Table 1). RTK: receptor tyrosine kinase; Mut: mutation; Amp: amplification.

**Table 1 cancers-13-05956-t001:** Ongoing clinical trials assessing combination therapies including covalent KRAS G12C inhibitors.

Drug	Sponsor	NCT Number	Trial Name	Phase	Disease or Condition	Combination Therapy *
AMG510 (sotorasib)	Amgen	NCT03600883	CodeBreaK 100	1/2	Advanced solid tumors	PD-1 or PD-L1 antibodies
		NCT04185883	CodeBreaK 101	1/2	Advanced solid tumors	AMG404 (PD-1 antibody)
						Trametinib (MEK inhibitor)
						RMC-4630 (SHP2 inhibitor)
						Afatinib (EGFR-TKI)
						Pembrolizumab (PD-1 antibody)
						Chemotherapy (CBDCA, PEM, DTX)
						Atezolizumab (PD-L1 antibody)
						Everolimus (mTOR inhibitor)
						Palbociclib (CDK4/6 inhibitor)
						MVASI^®^ (VEGF antibody)
						TNO155 (SHP2 inhibitor)
	Revolution Medicines	NCT05054725		2	Advanced NSCLC	RMC-4630 (SHP2 inhibitor)
MRTX849 (adagrasib)	Mirati	NCT03785249	KRYSTAL 1	1/2	Advanced solid tumors	Pembrolizumab (PD-1 antibody)
						Afatinib (EGFR-TKI)
		NCT04613596	KRYSTAL 7	2	Advanced NSCLC	Pembrolizumab (PD-1 antibody)
		NCT04975256	KRYSTAL 14	1	Advanced solid tumors	BI1701963 (SOS1 inhibitor)
GDC-6036	Roche/Genentech	NCT04449874		1	Advanced solid tumors	Atezolizumab (PD-L1 antibody)
						Bevacizumab (VEGF antibody)
						Erlotinib (EGFR-TKI)
						GDC-1971 (SHP2 inhibitor)
D-1553	InventisBio	NCT04585035		1/2	Advanced solid tumors	Standard treatment
JDQ443	Novartis	NCT04699188		1/2	Advanced solid tumors	TNO155 (SHP2 inhibitor)
						Spartalizumab (PD-1 antibody)
LY3537982	Eli Lilly	NCT04956640		1	Advanced solid tumors	Abemaciclib (CDK4/6 inhibitor)
						Erlotinib (EGFR-TKI)
						Sintilimab (PD-1 antibody)
						Temuterkib (ERK inhibitor)
						LY3295668 (Aurora kinase inhibitor)
						Cetuximab (EGFR antibody)
BI1823911	Boehringer	NCT04973163		1	Advanced solid tumors	BI1701963 (SOS1 inhibitor)

All clinical trials target *KRAS* G12C-mutated tumors. * Recruiting clinical trials of combination therapies for NSCLC are presented when identifiable. Information was obtained from the website: https://clinicaltrials.gov on 22 November 2021. EGFR-TKI: EGFR-tyrosine kinase inhibitor; CBDCA: carboplatin; PEM: pemetrexed; DTX: docetaxel.

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
