# Peer review of "Targeting Oncogenic KRAS in Non-Small-Cell Lung Cancer"

_cancers, 2021, doi:10.3390/cancers13235956_

Round 1

Reviewer 1 Report

Well-organized and informative review. Figures are informative. Reporting on results of clinical trials of KRAS G12C targeted therapies and their resistance mechanisms if timely.

A couple of suggestions for clarifications:

  1. Line 129: Consider adding "the PD-1 antibody" in front of nivolumab
  2. Lines 311-313: Please check this sentence for accuracy:

    In mutant KRAS-driven murine models of NSCLC, genetic deletion of PTPN11 suppressed tumor development, while dual inhibition of SHP2 and MEK resulted in sustained tumor growth in murine and human patient-derived organoids and xenograft models of NSCLC [96]

    From the reference 96 abstract  

    "...resulting in sustained tumor growth control in murine and human patient-derived organoids and xenograft models  of pancreatic ductal adenocarcinoma and non-small-cell lung cancer."

    1. Ruess, D.A.; Heynen, G.J.; Ciecielski, K.J.; Ai, J.; Berninger, A.; Kabacaoglu, D.; Gorgulu, K.; Dantes, Z.; Wormann, S.M.; Diakopoulos, K.N., et al. Mutant KRAS-driven cancers depend on PTPN11/SHP2 phosphatase. Nat Med 2018, 24, 954-960, doi:10.1038/s41591-018-0024-8.

Author Response

Thank you very much for reviewing our manuscript and your helpful suggestions.

Q1. Line 129: Consider adding "the PD-1 antibody" in front of nivolumab.

RE: We added "the PD-1 antibody" in front of nivolumab (page 4, line 142).

Q2. Lines 311-313: Please check this sentence for accuracy:  In mutant KRAS-driven murine models of NSCLC, genetic deletion of PTPN11 suppressed tumor development, while dual inhibition of SHP2 and MEK resulted in sustained tumor growth in murine and human patient-derived organoids and xenograft models of NSCLC [96]

From the reference 96 abstract

"...resulting in sustained tumor growth control in murine and human patient-derived organoids and xenograft models of pancreatic ductal adenocarcinoma and non-small-cell lung cancer."

96. Ruess, D.A.; Heynen, G.J.; Ciecielski, K.J.; Ai, J.; Berninger, A.; Kabacaoglu, D.; Gorgulu, K.; Dantes, Z.; Wormann, S.M.; Diakopoulos, K.N., et al. Mutant KRAS-driven cancers depend on PTPN11/SHP2 phosphatase. Nat Med 2018, 24, 954-960, doi:10.1038/s41591-018-0024-8.

RE: We revised the sentence to describe it accurately (page 9, lines 341-344).

Reviewer 2 Report

In the present manuscript, Noriaki Sunaga et al. reviewed genomic feathers and TME characteristics, targeted therapy, resistance mechanisms and progress of combined  therapy of KRAS-mutant NSCLC. The authors mainly focused on progress of targeted therapy, especially on the mechanism of resistance to targeted therapy. Finally, authors

Presented the possibility of combination therapy with targeted therapy , and suggested  combination medication may be a promising way to solve the problem faced by the clinical treatment of patients with KRAS mutant NSCLC. Thus, this manuscript provides an update on the most recent advances, and systematically summarizes the role of the common mutant gene KRAS in NSCLC, as well as targeted therapy, which could be useful for the further study of this field. In my opinion, this manuscript meets publication requirements. But several problems should be corrected to further increase the value of the present manuscript:

1. The part of “Genetic alterations of KRAS in human cancers” can be appropriately simplified to highlight the focus of the article, which is the treatment strategy of KRAS-mutant NSCLC.

2. Kras G12C inhibitors do not provide as durable responses as tyrosine kinase inhibitors against EGFR ALK, etc (months with KRAS G12C vs years with EGFR TKIs). This should be discussed.

3. KRAS targeted therapy were approved as the second line therapy after chemotherapy. However, the standard first line therapy strategies were PD-1 antibody plus chemotherapy. Thus, treatment sequence of the immunotherapy and targeted therapy should be discussed in whole process management of KRAS-mutated NSCLC.

4. Co-occuring tumor suppressor mutations include KEAP1 mutations, this needs to be included

5. In the part “Oncogenic KRAS regulates the tumor microenvironment (TME)” , the effect of KRAS only on the TME or the co-mutation with TP53, LKB1or KEAP1should be discussed separated, because the distinct TME were described previous.

6. In line 110, “KEAP1 mutations indicate these tumors commonly” , please confirm KEAP1 or LKB1 mutation indicate……

Author Response

Thank you very much for reviewing our manuscript. We agree with all of your insightful comments that enhance the quality of the manuscript.

Q1. The part of “Genetic alterations of KRAS in human cancers” can be appropriately simplified to highlight the focus of the article, which is the treatment strategy of KRAS-mutant NSCLC.

RE: We condensed the second part “Genetic alterations of KRAS in human cancers” by deleting the first paragraph “RAS proteins are small guanosine...”. Instead, we moved it to the part “Introduction” (page 1, line 43 to page 2, line 53), because we think it is critical to describe the general description on RAS proteins.

Q2. Kras G12C inhibitors do not provide as durable responses as tyrosine kinase inhibitors against EGFR ALK, etc (months with KRAS G12C vs years with EGFR TKIs). This should be discussed.

RE: We added a sentence to discuss this point (page 2, lines 59-62).

Q3. KRAS targeted therapy were approved as the second line therapy after chemotherapy. However, the standard first line therapy strategies were PD-1 antibody plus chemotherapy. Thus, treatment sequence of the immunotherapy and targeted therapy should be discussed in whole process management of KRAS-mutated NSCLC.

RE: We added sentences to describe this point (page 5, lines 215-220).

Q4. Co-occuring tumor suppressor mutations include KEAP1 mutations, this needs to be included.

RE: We added sentences to describe the critical issues regarding co-occurring KEAP1 mutations in KRAS-mutated NSCLC (page 3, lines 123-129).

Q5. In the part “Oncogenic KRAS regulates the tumor microenvironment (TME)” , the effect of KRAS only on the TME or the co-mutation with TP53, LKB1or KEAP1 should be discussed separated, because the distinct TME were described previous.

RE: We added a paragraph in the part of “Oncogenic KRAS regulates the tumor microenvironment (TME)” to describe the significance of co-occurring mutations in TP53, LKB1or KEAP1 on the TME in KRAS-mutated NSCLC (page 4, lines 166-176).

Q6. In line 110, “KEAP1 mutations indicate these tumors commonly”, please confirm KEAP1 or LKB1 mutation indicate……

RE: We revised the sentence (page 3, lines 118-120) to accurately describe the issue based on the reference by Skoulidis et al. (Cancer Discov 2015; 5: 860-877).

Reviewer 3 Report

The review titled "Targeting oncogenic KRAS in non-small cell lung cancer" is well written and exhaustive. The only comment I have is to add "disease or condition" to table 1. This update will enable an overall comparison between the drugs and trials.   

Author Response

Thank you very much for reviewing our manuscript and your helpful comments.

Q1. The only comment I have is to add "disease or condition" to table 1. This update will enable an overall comparison between the drugs and trials.  

RE: we revised Table 1 (page 7, line 291 to page 8, line 292) by adding a new column “disease or condition”. Since we found that current status of the KRYSTAL 2 trial (NCT04330664) has become “active, not recruiting” on November 22, 2021 (https://clinicaltrials.gov/ct2/results?cond=&term=NCT04330664&cntry=&state=&city=&dist=), we deleted this trial from the list of Table 1. We also revised the footnote as follows: “All clinical trials target KRAS G12C-mutated tumors. *Recruiting clinical trials of combination therapies for NSCLC are presented when identifiable. Information was obtained from the website: https://clinicaltrials.gov on November 22, 2021.”.